Network anomaly detection using Deep Autoencoder and parallel Artificial Bee Colony algorithm-trained neural network

Hacılar Hilal hilal.hacilar@agu.edu.tr 1
Dedeturk Bilge Kagan 2
Bakir-Gungor Burcu 1
Gungor Vehbi Cagri 3
1 Department of Computer Engineering, Abdullah Gul University , Kayseri , Turkey
2 Erciyes University, Department of Software Engineering , Kayseri , Turkey
3 Turkcell , Istanbul , Turkey
Sadek Rowayda
Electronic publication date: 2024 Oct 8
Publication date: 2024
Volume: 10
Electronic Location ID: e2333
Received 2023 Dec 21; Accepted 2024 Aug 25
Copyright: ©2024 Hacılar et al.
Copyright year: 2024
Copyright holder: Hacılar et al.
License: This is an open access article distributed under the terms of the Creative Commons Attribution License, which permits unrestricted use, distribution, reproduction and adaptation in any medium and for any purpose provided that it is properly attributed. For attribution, the original author(s), title, publication source (PeerJ Computer Science) and either DOI or URL of the article must be cited.
License URL: https://creativecommons.org/licenses/by/4.0/

Keywords: Artificial neural network, Artificial bee colony, Metaheuristics, Deep Autoencoder, Network intrusion detection systems (NIDS), Anomaly detection, UNSW-NB15, Swarm intelligence, NF-UNSW-NB15-v2

Funding: The authors received no funding for this work.

==============================
Cyberattacks are increasingly becoming more complex, which makes intrusion detection extremely difficult. Several intrusion detection approaches have been developed in the literature and utilized to tackle computer security intrusions. Implementing machine learning and deep learning models for network intrusion detection has been a topic of active research in cybersecurity. In this study, artificial neural networks (ANNs), a type of machine learning algorithm, are employed to determine optimal network weight sets during the training phase. Conventional training algorithms, such as back-propagation, may encounter challenges in optimization due to being entrapped within local minima during the iterative optimization process; global search strategies can be slow at locating global minima, and they may suffer from a low detection rate. In the ANN training, the Artificial Bee Colony (ABC) algorithm enables the avoidance of local minimum solutions by conducting a high-performance search in the solution space but it needs some modifications. To address these challenges, this work suggests a Deep Autoencoder (DAE)-based, vectorized, and parallelized ABC algorithm for training feed-forward artificial neural networks, which is tested on the UNSW-NB15 and NF-UNSW-NB15-v2 datasets. Our experimental results demonstrate that the proposed DAE-based parallel ABC-ANN outperforms existing metaheuristics, showing notable improvements in network intrusion detection. The experimental results reveal a notable improvement in network intrusion detection through this proposed approach, exhibiting an increase in detection rate (DR) by 0.76 to 0.81 and a reduction in false alarm rate (FAR) by 0.016 to 0.005 compared to the ANN-BP algorithm on the UNSW-NB15 dataset. Furthermore, there is a reduction in FAR by 0.006 to 0.0003 compared to the ANN-BP algorithm on the NF-UNSW-NB15-v2 dataset. These findings underscore the effectiveness of our proposed approach in enhancing network security against network intrusions.

Introduction

Advancements in intelligent technologies have significantly increased the number of internet users and applications. However, this rise in internet usage has also brought serious security challenges. According to the Cisco Cybersecurity Threat Trends Report (Cisco, 2023), in the previous year, cybercriminals launched an increased number of cyberattacks that were not only highly coordinated but also more advanced than ever before. These attacks aim to access sensitive data, steal credit card credentials, and disrupt information services. To address these vulnerabilities, various security mechanisms like firewalls, data encryption, and user authentication are implemented. Despite their effectiveness, they lack comprehensive packet analysis, making it challenging to detect all types of attacks. As a solution, network intrusion detection systems (NIDS) have been developed. NIDS continuously monitors networks for malicious activities and promptly alerts users to any intrusions or attacks, serving as an important line of defense against network threats.

Intrusion detection systems (IDS) and machine learning (ML) are two powerful technologies that are often combined to enhance the security of computer networks and systems. Artificial neural networks (ANNs) are a type of machine learning algorithm inspired by biological neural networks that mimic the learning and processing abilities of humans. The concept of ANN was initially introduced by McCulloch & Pitts (1943) in 1943, and since then, ANNs have been widely applied in various applications due to their capability for non-linear parameter mapping. ANNs are highly effective at modeling non-linear relationships. However, designing an appropriate network structure and finding ideal weight values pose significant optimization challenges. The conventional approach for training ANNs involves error back-propagation and weight adjustment using gradient descent (GD)-based algorithms. Due to the dependence of error surfaces on initial weights and parameters, these algorithms frequently get stuck in local minima.

Numerous studies in the literature have demonstrated successful outcomes by integrating various metaheuristic approaches with ANNs. Although the Artificial Bee Colony (ABC) algorithm is one of the most highly successful metaheuristic algorithms, a prevalent issue with this algorithm is its extended training time, leading many studies in the literature to focus on small datasets. Studies that employ the ABC-ANN algorithm with large datasets often lack detailed information regarding training time. To address this challenge, this study proposes a novel approach, i.e., a vectorized and parallelized Deep Autoencoder (DAE)-based hybrid ABC-ANN algorithm for binary classification tasks. This methodology leverages the respective strengths of DAE, the ABC algorithm, and parallel computing techniques to expedite the training process. In this respect, our study shows that ABC algorithm, with some modifications, can avoid local minimum solutions by conducting a high-performance search in the solution space.

In this study, our main contributions can be summarized as follows:

• As the sizes of storable and actively usable data continue to increase every day, the importance of GPU parallelization cannot be overlooked. Despite the ABC algorithm being a highly successful metaheuristic algorithm, there are deficiencies and gaps concerning its applicability. Table 1 summarizes studies that apply ABC-ANN and other metaheuristics, highlighting gaps such as applicability to large datasets and the lack of information about training times even when large datasets are used. This study develops an ABC-based ANN model that is simplified to work efficiently on modern hardware, reducing computational complexity as much as possible (vectorization) and making it suitable for GPU execution (parallelization). The model is tailored for binary classification tasks.

• This study introduces a novel approach by proposing a DAE-based, vectorized, and parallelized ABC-ANN algorithm for binary classification. The aim is to enhance classification accuracy and detection rate. By employing the DAE, the algorithm extracts relevant and distinctive features from the input data, leading to a more effective detection process.

• An XGBoost-based feature selection approach has been implemented to reduce significant computational costs associated with typical ANN-based models using ABC algorithms. This technique effectively decreases the number of dimensions in the input data, therefore reducing the computing cost.

• Different evaluation metrics, such as accuracy, f1-measure, detection rate, false alarm rate, and training time, were used to test and compare how well the proposed method works with existing machine learning techniques. This detailed assessment offers valuable insights regarding the efficiency of the suggested approach.

• To automatically optimize the hyperparameters of the proposed ABC-ANN approach and the metaheuristics, the Bayesian parameter optimization method is utilized. This optimization method intelligently explores the hyperparameter space, facilitating the selection of the best hyperparameter configurations for each model.

• By incorporating these advancements, the proposed approach outperforms some metaheuristics in terms of precision, f1-measure, detection rate, false alarm rate, and training time.

The rest of this paper is organized as follows: ‘Related Work’ provides an overview of related work, encompassing topics such as ABC-ANN, NIDS, and metaheuristics on NIDS. ‘Proposed Method’ offers a detailed explanation of the proposed DAE-based parallel ABC-ANN method. Experimental results on two datasets, along with discussions, are presented in ‘Experimental Results’. Lastly, ‘Conclusion’ presents the conclusions of this study.

Table 1 Related works that apply ABC-ANN and other metaheuristics.

Reference	Problem	Method	Optimized components	Data size	ttime	
Ali, Dewangan & Dewangan (2018)	DDos attack detection	ABC-ANN	weights	X	X	
Karaboga & Akay (2007)	XOR, 3-Bit Parity and 4-Bit Encoder-Decoder	ABC-ANN	weights	X	X	
Ozturk & Karaboga (2011)	XOR, 3-Bit Parity and 4-Bit Encoder-Decoder	ABC-LM-ANN	weights	X	X	
Zhou et al. (2020)	prediction of the heating and cooling loads of residential buildings	ABC-MLP PSO-MLP	weights	8 features 768 samples	X	
Jahangir & Eidgahee (2021)	FRP-concrete bond strength evaluation	ABC-ANN	weights	656 samples	X	
Taheri et al. (2017)	forecasting the blast-produce ground vibration	ABC-ANN	weights	89 samples	X	
Hajimirzaei & Navimipour (2019)	intrusion detection for cloud computing	ABC-ANN	weights	41 features 7 million samples	X	
Ghanem et al. (2020)	network intrusion detection	ABC-DA-ANN	weights	(big data) UNSW-NB15, ISCX2012 KDD Cup 99, NSL-KDD	X	
Karuppusamy et al. (2022)	network intrusion detection	SSA-DBN	weights	(big data) KDD cup, BoT-IoT	✓	
Ahmad et al. (2022)	IIoT network intrusion detection	PSO-SQP, RaNN	hyperparameters	(big data) DS2OS, UNSW-NB15, ToN_IoT	X	
Elmasry, Akbulut & Zaim (2020)	network intrusion detection	PSO, LSTM-RNN, DNN and DBN	no. of features and hyperparameters	(big data) CICIDS201 and NSL-KDD	✓	
Kanna & Santhi (2022)	network intrusion detection	ABC , BWO CNN, LSTM	no. of features and hyperparameters	(big data) NSL-KDD, ISCX-IDS UNSW-NB15, CSE-CIC-IDS2018	✓	
Saif et al. (2022)	network intrusion detection	ABC-DA-ANN	no. of features	(big data) NSL-KDD	✓	
Ghanbarzadeh, Hosseinalipour & Ghaffari (2023)	network intrusion detection	HOA	no. of features	(big data) NSL-KDD and CSE-CIC-IDS2018	X	
Malibari et al. (2022)	network intrusion detection	QPSO, DWNN	hyperparameters	(big data) CICIDS2017	X	
Ponmalar & Dhanakoti (2022)	network intrusion detection	WOA-Tabu CNN	hyper parameters	(big data) NSL-KDD, KDD-Cup99, UNSW-NB15	X	
Proposed method	network intrusion detection	DAE-ABC-ANN	weights	(big data) UNSW-NB15 and NF-UNSW-NB15-v2	✓	
Notes.

ttime, training time.

Related Work

To enhance the clarity of the literature review in this study, we have organized it into three sub-sections.

ABC-ANN literature review

The ABC technique is utilized to estimate the weight and bias values of the neural network model by minimizing the mean square error between the target and the output of the ANN. Numerous studies in the literature utilize ANNs to address a wide range of problems, including the field of IDS.

Ali, Dewangan & Dewangan (2018) employ ABC for both feature selection and ANN weight optimization in order to detect DDoS attacks. They use a back-propagation neural network architecture that feeds inputs and adjusts weights simultaneously. However, there is no implementation and the performance evaluation metrics are not provided for any dataset.

Karaboga & Akay (2007) suggests training an ANN using ABC and comparing its performance with other population-based algorithms. Ozturk & Karaboga (2011) proposes a hybrid model that combines ABC and Levenberg–Marquardt (LM) algorithms for training an ANN model. Both studies evaluate their models using XOR, 3-Bit Parity, and 4-bit Encoder-Decoder problems. They highlighted the potential of using the ABC algorithm as an optimization technique for training ANNs. In training the ANN, in agreement with earlier studies, Ozkan, Kisi & Akay (2011) showed that the ABC approach outperformed the back-propagation algorithm.

Zhou et al. (2020) combine particle swarm optimization (PSO) and ABC algorithms to optimize the weights of a multi layer perceptron (MLP) for predicting the heating and cooling loads of residential buildings using 768 samples. Anuar, Selamat & Sallehuddin (2015) proposes the ABC-ANN method for crime classification using a crime dataset with 128 attributes and 1,994 instances. They evaluate the performance of ANN-ABC using only the accuracy metric.

Other related studies include ANN trained by ABC for FRP-concrete bond strength evaluation (Jahangir & Eidgahee, 2021) using 656 samples, forecasting blast-produced ground vibration (Taheri et al., 2017) with 89 blasting events, determining the vibration period of reinforced concrete infilled framed structures (Asteris & Nikoo, 2019) with 4,025 samples, and intrusion detection using a combination of fuzzy clustering, MLP, and ABC (Hajimirzaei & Navimipour, 2019) on the NSL-KDD and CloudSim datasets.

Mahmod, Alnaish & Al-Hadi (2015) used the ABC-ANN model for intrusion detection and achieved 87% accuracy on the NSL-KDD dataset. However, their study did not focus on time and speed considerations or the use of a hybrid approach combining the Deep Autoencoder and the ABC-ANN model.

In summary, the ABC algorithm is used in literature for training ANN models to avoid local minimum solutions. However, it suffers from long training times to find global solutions. Existing studies that use the ABC approach for ANN training are often trained on small datasets. To address these challenges, this study proposes a novel hybrid approach combining Deep Autoencoder and ANN models trained by a parallel Artificial Bee Colony algorithm with Bayesian hyperparameter optimization.

Network intrusion detection systems (NIDS) literature review

Anomaly detection, especially in NIDS, has remained a long-standing yet dynamically evolving research domain across various research communities for decades (Pang et al., 2021). Most studies utilize machine learning and deep learning techniques, and hybridize them with various techniques such as fuzzy logic-based decision systems (Javaheri et al., 2023), to detect network anomalies.

Some studies employ different concepts to detect anomalies in network traffic data. One of these studies, Jain, Kaur & Saxena (2022), uses concept drift to detect attacks in network flows by monitoring changes in the network traffic distribution or alterations in the characteristics of the network traffic. They use the support vector machine (SVM) algorithm for classification and obtain satisfactory performance metrics on Testbed Dataset, NSL-KDD and CIDDS-2017.

Zhong et al. (2020) introduces a novel anomaly detection framework that integrates multiple deep learning techniques, including the Damped Incremental Statistics algorithm for feature extraction from network traffic, the Autoencoder for assigning abnormal scores to network traffic, LSTM for classification, and a weighted method for obtaining the ultimate abnormal score. Analyzing the mirai dataset (Mirsky et al., 2018), the authors show that the HELAD algorithm demonstrates good adaptability and accuracy compared to other state-of-the-art algorithms. Another study (Chen et al., 2022) comprises two primary steps. Firstly, a Deep Belief Network (DBN) is employed for nonlinear feature extraction, automatically extracting features from the data while reducing its dimensionality. Subsequently, a lightweight long short-term memory (LSTM) network is utilized to classify the extracted features, thereby generating classification results. The researchers tested their model on the KDD99 and CICIDS2017 benchmark datasets, obtaining satisfactory results.

Detecting abnormal patterns and attacks using graph-based anomaly detection is another area of focus. In a study conducted by Deng & Hooi (2021), a graph deviation network (GDN) approach based on graph neural networks (GNNs) has been proposed, yielding significant results in detecting anomalies and attacks in the sensor data of cyber-physical systems. In another study, Ding et al. (2021) address the issue of few-shot network anomaly detection by proposing a novel family of graph neural networks called GDN. These networks can utilize a limited number of labeled anomalies to enforce statistically significant deviations between abnormal and normal nodes in a network.

Metaheuristics on NIDS

In literature, metaheuristics are commonly used for different objectives, such as feature selection (Najafi Mohsenabad & Tut, 2024; Sanju, 2023; Donkol et al., 2023) and hyperparameter optimization. Only a few researchers have utilized metaheuristics with the aim of training neural networks and deep learning architectures (Kaveh & Mesgari, 2022). Ghanem et al. (2020) has constructed an NIDS model for training MLP using a hybrid metaheuristic that combines the Artificial Bee Colony (ABC) algorithm and the Dragonfly Algorithm (DA). This study has obtained significant results in terms of DR, FAR, and accuracy on different public network datasets. Karuppusamy et al. (2022) has offered a method based the Chronological Salp Swarm Algorithm for the weight optimization of DBN for the detection of intrusions. They have performed experiments on the KDD cup and the BoT-IoT datasets and reported significant results.

Ahmad et al. (2022) has developed a RaNN model whose hyperparameters are tuned by hybrid PSO with sequential quadratic programming (SQP). In the study of Elmasry, Akbulut & Zaim (2020), they have utilized PSO for both the purposes of feature selection and hyperparameter optimization. Subsequently, they have tested this pre-trained model using three deep learning algorithms: DNN, LSTM-RNN, and DBN. Kanna & Santhi (2022), in the first stage, have applied feature selection by the ABC algorithm and hyperparameter optimization by Black Window Optimization (BWO) algorithms. Subsequently, they have applied Convolutional and LSTM neural networks to intrusion detection. Saif et al. (2022) has tried to reduce the computational cost of intrusion detection systems for IoT based healthcare systems via metaheuristic algorithms. They have employed algorithms such as PSO, Genetic Algorithm (GA), and differential evolution (DE), attaining substantial outcomes on the NSL-KDD dataset. Ghanbarzadeh, Hosseinalipour & Ghaffari (2023) has employed a novel approach called the Horse Herd Optimization Algorithm (HOA) that mimics horse behaviors within a herd to select relevant features for detecting intrusions. It has obtained significant results on the NSL-KDD and CSE-CIC-IDS2018 datasets. On the other hand, the study by Malibari et al. (2022) has employed a metaheuristic called quantum-behaved particle swarm optimization (QPSO) to optimize hyperparameters of the deep wavelet neural network (DWNN) model. This model is designed to construct intrusion detection systems for secure, smart environments. Ponmalar & Dhanakoti (2022) has optimized CNN hyperparameters via a hybrid metaheuristic approach, which is a combination of both the whale optimization algorithm and the local search of the Tabu optimization algorithm.

In spite of the fact that all studies have different contributions to network intrusion detection research, they may have some limitations, such as higher computational complexities, longer training times, and lower detection rates. This study suggests overcoming the above-mentioned limitations.

Proposed Method

This section comprehensively explores various aspects crucial to the development and implementation of an effective network intrusion detection system (NIDS). In this context, our threat model encompasses the following components:

• Assets: The elements within the network that need protection, including data, network infrastructure components (such as routers and switches), servers, and endpoints (such as computers and IoT devices).

• Threat actors: Malicious hackers, insider threats, or other entities attempting to compromise the network’s security.

• Attack vectors (Hindy et al., 2020): Network attacks including DoS, backdoors, generic attacks, analysis attacks, exploits, shell code, fuzzers, reconnaissance, and worms (Detailed in Table 2).

• Attack surface: Network protocols, communication channels, network devices, and endpoints.

• Security controls: These are the measures put in place to detect and mitigate anomalous behavior and potential security threats within the network. Security controls in our study include intrusion detection systems (IDS) and anomaly detection algorithms based on DAE and parallel ABC algorithms.

Along this line, Fig. 1 illustrates the workflow of our study. Beginning with feature extraction via Deep Autoencoder (DAE) and feature selection via the extreme gradient boosting (XGBoost) algorithm, these components handle the critical processes of data preprocessing and feature engineering. Subsequently, the Artificial Bee Colony (ABC) algorithm and its adaptation to the artificial neural network (ANN) framework are detailed, highlighting the innovative approach taken to optimize model training. Moreover, the section discusses the significance of data vectorization and parallel computation on GPUs, shedding light on the computational strategies employed to enhance efficiency and scalability. Lastly, it addresses the utilization of Bayesian optimization, offering insights into the techniques employed for fine-tuning model parameters and maximizing classification performance. Through an in-depth examination of these key components, this section proposes a DAE-based parallel ABC-ANN method, contributing to advancements in network intrusion detection methodologies.

Table 2 Attack types and their short descriptions in the UNSW-NB15 and the NF-UNSW-NB15-v2 datasets.

Type	Description	
Normal	Network traffic that is expected under regular operating conditions.	
Fuzzers	Attempting to cause a program or network to suspend or crash by feeding it randomly generated data (Thanh & Van Lang, 2020; Sarhan et al., 2023).	
Analysis	Examining network traffic patterns to gather sensitive information and infer activities without intercepting or decrypting the actual data. Includes attacks like port scans, spam, and HTML file penetrations (Moustafa & Slay, 2015; Dada et al., 2019).	
Backdoors	A technique that stealthily bypasses a system’s security mechanism to gain access to a computer or its data (Moustafa & Slay, 2015; Li et al., 2022).	
DoS	Aims to make a computer or network service unavailable to its intended users by overwhelming it with a flood of illegitimate requests or exploiting vulnerabilities to crash the system (Moustafa & Slay, 2015; Yuan, Li & Li, 2017; Douligeris & Mitrokotsa, 2004).	
Exploits	Methods or tools used by attackers to take advantage of vulnerabilities or flaws in software, hardware, or operating systems to gain unauthorized access or cause damage (Singh, Joshi & Kanellopoulos, 2019).	
Generic	A technique that targets all block ciphers with a specific block and key size, regardless of the block cipher’s internal structure (Moustafa & Slay, 2015; Alsariera, 2021).	
Reconnaissance	Includes all strikes capable of simulating information-gathering attacks (Uma & Padmavathi, 2013).	
Shellcode	Small piece of code used as the payload in exploiting software vulnerabilities, designed to grant the attacker control over the compromised system (Arce, 2004; Onotu, Day & Rodrigues, 2015).	
Worms	A self-replicating malware that spreads across networks by exploiting vulnerabilities, often without user intervention. It can cause harm by consuming bandwidth, overloading systems, and delivering payloads such as additional malware (Freund & Schapire, 1997).	

Figure 1 The workflow of the proposed network intrusion detection methodology, including the preprocessing, feature extraction and selection, model construction, and model evaluation processes highlighted in red, orange, blue, and green, respectively.

Feature extraction via Deep Autoencoder (DAE)

Deep Autoencoders (DAE) are a form of deep neural network, which is used to reduce dimensionality and extract attributes. The main purpose of a DAE is to discover a compressed representation of input data while minimizing information loss. This is done by training the network to reproduce the input in the output layer. Figure 2 shows an example of the architecture of a DAE consisting of an encoder that converts input data to a compressed version and a decoder that recreates the original input from the encoded data. Encoders compress the data into a lower-dimensional space and effectively capture the most important features of the input in a non-linear way. In this study, a new encoded representation of the input data is extracted using DAE.

Figure 2 Illustration of an example of Deep Autoencoder architecture.

This investigation utilized Bayesian optimization (‘Bayesian Optimization’). Hyperparameters were optimized for both datasets. Encoded data capturing the most important features of original data has been combined with original data for further analysis. This consolidated data set was then used as an input for the feature selection step.

By using a DAE for feature extraction, this study aims to obtain a more compact and informative representation of the data, which can potentially improve the performance of the following analysis tasks, such as classification or anomaly detection. The encoded features can help reduce the dimensionality of the data while retaining important information, thus aiding in more efficient and effective feature selection processes.

Feature selection via extreme gradient boosting algorithm

Ensembles of decision tree approaches, such as extreme gradient boosting (XGBoost), have the advantage of being able to automatically generate feature importance scores from a trained model. These scores indicate the relative importance or usefulness of each feature in the model’s decision-making process. Features that are consistently used to make crucial decisions in the ensemble of decision trees will have higher importance scores.

This study aims to find the optimal set of features from the UNSW-NB15 and the NF-UNSW-NB15-v2 datasets by considering both the original features and the encoded features obtained from the DAE. To achieve this, the feature importance scores provided by XGBoost are utilized. To calculate the feature importance scores and to select the most informative features, a five-fold cross-validation is employed. Then, the F1-scores and accuracy scores for different combinations of selected features, including both the original and encoded features, are examined.

Our preliminary analyses showed that the best results in terms of accuracy and F1-score are obtained when the encoded features and original features are concatenated. In the UNSW-NB15 dataset, only the top 30 features are selected based on the XGBoost feature importance scores, while in the NF-UNSW-NB15-v2 dataset, only 40 features are selected. Therefore, the further experiments are carried out using the subsets of 30 and 40 selected features, respectively. This approach allows us to focus on the most relevant features and potentially improve the accuracy of the findings.

Artificial Bee Colony algorithm

The Artificial Bee Colony (ABC) algorithm operates by simulating the behavior of honey bees. In the ABC algorithm, each food source represents a potential solution to the optimization problem, and the quantity of food source indicates the quality or fitness of the solution.

The ABC algorithm consists of three main phases: the employed bee phase, the onlooker bee phase, and the scout bee phase. These phases collectively form an iterative process to search for optimal solutions.

In the employed bee phase, there are the same number of employed bees as there are food sources. Each employed bee examines a new food source in the neighborhood of its current food source. If the quantity (fitness) of the new source is higher than that of the previous source, the employed bee updates its memory by recording the new food source and forgets the previous one. The employed bees then perform a dance within the colony to communicate the quantity and quality of their food sources.

In the onlooker bee phase, the onlooker bees observe the dance of the employed bees and choose their food sources based on the quality of the food source. The higher the quantity and quality of a food source, the more likely it is to be chosen by the onlooker bees. Each onlooker bee assesses a new food source in the neighborhood of the chosen food source, similar to what the employed bees do.

In the scout bee phase, abandoned food sources that have not been improved for a certain number of iterations are identified, indicating that they are not promising solutions. These abandoned food sources are replaced with new and unexplored food sources found by scout bees, which explore new areas of the search space.

These three stages are repeated iteratively until the termination criteria or requirements of the optimization problem are met. The ABC algorithm aims to find the optimal solution by continuously exploring the search space based on the information shared among employed bees, onlooker bees, and scout bees. Through this iterative process, the algorithm can efficiently search for high-quality solutions in the optimization problem domain.

Artificial neural network

Artificial neural network (ANN) models are computational systems inspired by the neural structure of the human brain, aiming to replicate the information processing mechanisms of biological systems. These models consist of interconnected nodes, called neurons, organized into layers, enabling them to learn and adapt from data. Data is received by the input layer and propagated through weighted connections to hidden layers, ultimately generating an output. Throughout the training process, the network adjusts these weights based on the provided dataset, enhancing its predictive or classification capabilities.

The effectiveness of ANNs lies in their ability to extract hierarchical features and perform nonlinear mapping, enabling them to capture intricate relationships within data. In this study, to address this capability, the ABC algorithm is employed during the training phase of ANNs. This integration aims to prevent the occurrence of local minima and explore high-performance solutions within the solution space, thereby enhancing the robustness and effectiveness of the ANN model for various optimization tasks.

Adaptation of ABC to ANN

In this study, the ABC algorithm, a population-based optimization technique, is customized for use with ANNs to optimize the weights and biases. The original ABC algorithm’s primary drawback is its long training times, especially when attempting to locate global solutions. To tackle this issue, this study suggests utilizing a vectorized and parallelized ABC-ANN algorithm. This proposed approach combines the advantages of the ABC algorithm with parallel computing techniques, effectively expediting the training process (‘Data Vectorization and Parallel Computation on GPU’).

The proposed neural network structure, as depicted in Fig. 3, consists of an input layer where each neuron represents a feature from the intrusion dataset along with a bias value. The sigmoid function (shown in Eq. (1)) is used for the activation of all weights. The hidden layer neurons and connections between the input and output layers imitate and simulate the structure of the human brain. Equation (2) calculates the values of the hidden layers to produce the probability value in the subsequent step. The output layer produces binary outcomes according to Eq. (3), where 0 is used for normal and 1 is used for attack, using the probability function shown in Eq. (4).

Figure 3 Illustration of proposed ANN architecture that follows a standard ANN structure.

The proposed parallel ABC-ANN algorithm ( Algorithm 1 ) combines the ABC optimization technique with ANN to create an effective classification method, termed as the ABC-ANN classification method. It combines the collective and global search intelligence of bees to optimize the weight parameters of an ANN for classification tasks. Initially, the algorithm calculates the total number of weights and biases using the formula (presented in line 1 of the Algorithm 1 ): D = (N + 1) ×HLS + (HLS + 1)) in the ANN model based on the number of neurons in the input and hidden layers, as well as the output layer. In the formula, N represents the number of neurons in the input layer and includes a bias term, while HLS denotes the number of neurons in the hidden layer. This sets the dimensionality of the solution space for the ABC algorithm.

Following the generation of the weight matrix, a matrix of food sources, defined by dimensions PxD, is generated for the solution space ( Algorithm 2 ). Here, P represents the number of food sources (solutions), and D refers to the dimensionality of each solution. The fitness values of these solutions are computed based on their performance in the classification task (line 4).

In this study, two versions of the fitness function were implemented, taking into account the necessary adjustments for optimization across two different network datasets. The UNSW-NB15 dataset has a low class imbalance ratio; thus the mean absolute error (MAE) ( Algorithm 3 ) was optimized. On the other hand, the NF-UNSW-NB15-v2 dataset exhibits a significantly high class imbalance ratio; therefore the F1-score ( Algorithm 4 ) was prioritized for optimization.

Taking into account the classification results from the output layer ( Algorithm 5 ), F1 or accuracy scores are calculated for each food source, and then the fitness values are computed by averaging them.

Employed bees perform local searches around food sources ( Algorithm 7 ), and if a new solution offers improvement, it replaces the old one. Onlooker bees select food sources based on their fitness values ( Algorithm 8 ), conducting searches preferentially around better-performing solutions. If a solution’s limit counter exceeds a predefined threshold, indicating no improvement, a scout bee generates a new solution ( Algorithm 9 ). The best solution found represents the optimal or near-optimal set of weights for the ANN ( Algorithm 10 ). The algorithm iterates over the search process until the maximum number of evaluations (MEN) is reached (lines 7-31 in Algorithm 1 ).

By combining the exploration capabilities of the ABC algorithm with the learning and optimization capabilities of ANN, the proposed parallel ABC-ANN algorithm aims to find optimal weight values for the neural network to accurately detect network anomalies.

(1) σx=11+e−x

(2) hi=σx1w1i′+x2w2i′+…+xnwni′+wbi′

(3) y=σh1w11′′+h2w21′′+…+hmwm1′′+wb1′′

(4) p=1,ify≥0.50,otherwise.

___________________________________________________________ Algorithm 1 Proposed ABC-ANN classification method_____________________________________________________________________________________________________________________________________________________________________________________________________ 1: Determine the input parameters: Input matrix XM×N, target → yM, number of food sources P, position of the food sources WP×D, maximum evaluation number MEN, lower bound lb, upper bound ub, limit, modification rate MR, hidden layer size HLS Output:  1:  D ← (N + 1) × HLS + (HLS + 1)   2:  GENERATE_FOOD_SOURCES()   3:  W′  ← W  4:   → fit ← CALC_FIT(W)   5:  → τ ← zeros(P)                                                                                                                                                                                                                    ⊳ P-dimensional zero vector   6:  evaluation_number ← 0   7:  while evaluation_number < MEN do  8:       SEND_EMPLOY ED_BEES()   9:         → sfit ← CALC_FIT(W′  ) 10:        → ind ← → sfit >  → fit 11:         → rind ← → sfit ≤ → fit 12:       → τ[ →ind] ← 0 13:       W[ →ind] ← W′  [ →ind] 14:        → fit[ →ind] ← → sfit[ →ind] 15:       → τ[  →rind] ← → τ[  →rind] + 1 16:       CALC_PROBABILITIES() 17:       SEND_ONLOOKER_BEES() 18:         → sfit ← CALC_FIT(W′  ) 19:       for i ← 1 : P do 20:            t ←  → tmpID[i] 21:            if   → sfit[i] >  → fit[t] then 22:                  → τ[t] ← 0 23:                  W[t,:] ← W′  [i,:] 24:                   → fit[t] ← → sfit[i] 25:            else 26:                  → τ[t] ← → τ[t] + 1 27:            end if 28:       end for 29:       SEND_SCOUT_BEES() 30:       MEMORIZE_BEST_SOURCE() 31:  end while 32:  return    → gpar                                                                                                                                                                                          ⊳ return global params

________________________________________________________________________________________________________________________________________________________________________________________________________________________________________________________________________________ Algorithm 2 Create Food Source Positions______________________________________________________________________________________________________________________________________________________________________________________________________________________   1:  procedure GENERATE_FOOD_SOURCES  2:       for i ← 1 : P do  3:            for j ← 1 : D do  4:                  W[i,j] ← lb + rand(0,1) × (ub − lb)   5:            end for  6:       end for  7:  end procedure_____________________________________________________________________________________________________________________________________________________________________________________________________________________

____________________________________________________________________________________________________________________________________________________________________________________________________________________________________________ Algorithm 3 Calculate Mean Absolute Error based Fitness Function____________________________________________________________________________________________________________________________________________________________________________________   1:  procedure CALC_FIT(ϕ)   2:       ps ← CALCOutputLayer(ϕ)   3:       ϵ ← absolute(ps − → yM)   4:       f ← mean(ϵ,axis = 0)   5:       evaNumber ← evaNumber + len(f)   6:       return f/(1 + f)   7:  end procedure_____________________________________________________________________________________________________________________________________________________________________________________________________________________

____________________________________________________________________________________________________________________________________________________________________________________________________________________________________________ Algorithm 4 Calculate F1-score based Fitness Function_____________________________________________________________________________________________________________________________________________________________________________________________________   1:  procedure CALC_FIT(ϕ)   2:       ps ← CALCOutputLayer(ϕ)   3:       ps ← round(ps)   4:       f ← F1_score(→yM,ps)   5:       evaNumber ← evaNumber + len(f)   6:       return f  7:  end procedure_____________________________________________________________________________________________________________________________________________________________________________________________________________________

____________________________________________________________________________________________________________________________________________________________________________________________________________________________________________ Algorithm 5 Calculate Output Layer_____________________________________________________________________________________________________________________________________________________________________________________________________________________________   1:  procedure CALCOutputLayer(ϕ)   2:       M,N ← X.shape  3:       P ← ϕ.shape[0]   4:       ps ← zeros(M,P)                                                                                                                                                                                                                        ⊳ output neurons   5:       ps ← ps + ϕ[:,−1]                                                                                                                                                                                                                           ⊳ bias addition   6:       for i ← 0 : HLS do  7:            W ← ϕ[:,i × N : (i + 1) × N]T   8:            b ← ϕ[:,N × HLS + HLS + i]T   9:            zi ← σ(X.dot(W) + b)                                                                                                                                                                                                            ⊳ σ is sigmoid func 10:            ps ← ps + zi ∗ ϕ[:,FV S × HLS + i] 11:       end for 12:       ps ← σ(ps) 13:       return ps 14:  end procedure_____________________________________________________________________________________________________________________________________________________________________________________________________________________

____________________________________________________________________________________________________________________________________________________________________________________________________________________________________________ Algorithm 6 Calculate Probabilities_______________________________________________________________________________________________________________________________________________________________________________________________________________________________   1:  procedure CALC_PROBABILITIES  2:       maxfit ← max( →fit)   3:       prob ← (0.9 × ( →fit/maxfit)) + 0.1   4:  end procedure_____________________________________________________________________________________________________________________________________________________________________________________________________________________

____________________________________________________________________________________________________________________________________________________________________________________________________________________________________________ Algorithm 7 Employed Bee Phase_________________________________________________________________________________________________________________________________________________________________________________________________________________________________   1:  procedure SEND_EMPLOYED_BEES  2:       for i ← 1 : P do  3:             → ar ← rand(low = 0,high = 1,size = (D))   4:            → ρ ← → ar < MR                                                                                                                                                                                      ⊳ param to change   5:            η ← randint(1,P), η ⁄= i                                                                                                                                                                          ⊳ choose neighbour   6:            W′  [i,:] ← W[i,:]   7:            vec ← W′  [i,→ ρ]   8:            vec ← vec + rand(−1,1) × (vec − W[η,→ ρ])   9:            vec[vec < lb] ← lb 10:            vec[vec > ub] ← ub 11:            W′  [i,→ ρ] ← vec 12:       end for 13:  end procedure_____________________________________________________________________________________________________________________________________________________________________________________________________________________

____________________________________________________________________________________________________________________________________________________________________________________________________________________________________________ Algorithm 8 Onlooker Bee Phase__________________________________________________________________________________________________________________________________________________________________________________________________________________________________   1:  procedure SEND_ONLOOKER_BEES  2:       i ← 0   3:       t ← 0   4:       while t < P do  5:            if rand(0,1) < prob[i] then  6:                  → ar ← rand(low = 0,high = 1,size = (D))   7:                  → ρ ← → ar < MR                                                                                                                                                                                  ⊳ param to change   8:                  η ← randint(1,P), η ⁄= i                                                                                                                                                                              ⊳ neighbour   9:                  W′  [t,:] ← W[i,:] 10:                  vec ← W′  [t,ρ] 11:                  vec ← vec + rand(−1,1) × (vec − W[η,→ ρ]) 12:                      → tmpID[t] ← i 13:                  vec[vec < lb] ← lb 14:                  vec[vec > ub] ← ub 15:                  W′  [t,→ ρ] ← vec 16:                  t ← t + 1 17:            end if 18:            i ← i + 1 19:            if i ≥ P then 20:                  i ← 0 21:            end if 22:       end while 23:  end procedure_____________________________________________________________________________________________________________________________________________________________________________________________________________________

____________________________________________________________________________________________________________________________________________________________________________________________________________________________________________ Algorithm 9 Scout Bee Phase_______________________________________________________________________________________________________________________________________________________________________________________________________________________________________   1:  procedure SEND_SCOUT_BEES  2:       index ← argmax(→τ)   3:       if → τ[index] ≥ limit then  4:            for j ← 1 : D do  5:                  W[index,j] ← lb + rand(0,1) × (ub − lb)   6:                  W′  [index,j] ← W[index,j]   7:            end for  8:            fit[index] ← CALC_FIT(W[index,:])   9:            → τ[index] ← 0 10:       end if 11:  end procedure_____________________________________________________________________________________________________________________________________________________________________________________________________________________

____________________________________________________________________________________________________________________________________________________________________________________________________________________________________________ Algorithm 10 Memorize Best Source_____________________________________________________________________________________________________________________________________________________________________________________________________________________________   1:  procedure MEMORIZE_BEST_SOURCE  2:       index ← argmax( →fit)   3:       if  → fit[index] > gmax then  4:            gmax ← → fit[index]                                                                                                                                                                                                                ⊳ global maximum   5:               → gpar ← W[index,:]                                                                                                                                                                                                                    ⊳ global params   6:       end if  7:  end procedure_____________________________________________________________________________________________________________________________________________________________________________________________________________________

Data vectorization and parallel computation on GPU

The ABC-ANN algorithm requires a robust acceleration mechanism to effectively handle big data challenges and achieve faster convergence to a global solution. To address this need, the vectorization and GPU parallelization have been employed to enhance the computational efficiency of the optimization process.

Vectorization involves transforming mathematical operations into vector form, leveraging the computational capabilities of modern processors for parallel execution. By leveraging the NumPy library, which is widely used for numerical computing in Python, the code is designed to perform array operations efficiently and in parallel. The use of vectorized operations in NumPy eliminates the need for explicit looping and indexing, allowing shorter and more readable code. This method not only reduces the number of lines of code but also reduces the possibility of occurrence of bugs and errors. Additionally, vectorization provides significant performance enhancements. Utilizing the underlying C implementation of NumPy enables efficient parallel execution of array operations. This provides faster execution times compared to sequential processing, where traditional loops are used. The main benefits of vectorized code are enhanced readability, decreased code complexity, and increased computational efficiency. It allows for code that is cleaner, making it simpler to understand and maintain. Moreover, parallel execution of operations can result in significant performance improvements, particularly when dealing with large datasets or problems with high computational costs.

With the rapid advancement of GPU technologies, researchers are increasingly turning to parallel computing to boost algorithm speed. In this regard, the CuPy library for Python, developed by Nishino & Loomis (2017), has gained prominence. CuPy is an open-source Python library designed to harness NVIDIA GPUs to accelerate matrix operations. It is fully compatible with NumPy and enables the utilization of modern GPU capabilities through a compatible interface.

In this study, all data used in the training phase of the ABC-ANN algorithm were condensed into minimal matrices and converted into first NumPy and then Cupy arrays to optimize calculation speed. Overall, the utilization of vectorization and GPU parallelization via the CuPy library in the ABC optimization code of this study enhances the efficiency and readability of the implementation, rendering it a valuable tool for scientific computing and optimization tasks.

Bayesian optimization

Bayesian optimization is a technique that leverages Bayes’ theorem to efficiently search for the global optimum of an objective function. It involves constructing a probabilistic model, known as the surrogate function, which represents the objective function. This surrogate function is then iteratively evaluated and updated based on the observed results.

In the context of machine learning, Bayesian optimization is commonly used for hyperparameter tuning. Hyperparameters are configuration settings of a model that are not learned from the data but need to be specified by the user. Finding the optimal combination of hyperparameters is crucial for achieving the best performance of a machine learning model on a given dataset. Hyperparameter tuning is a challenging task as it involves searching through a large space of possible hyperparameter values. The objective function, which is typically the performance metric of the model on a validation set, is often complex and computationally expensive to evaluate.

Bayesian optimization provides a systematic approach to efficiently search for the optimal hyperparameters. Unlike random or grid search methods, Bayesian optimization maintains a record of previous evaluation results. These results are utilized to construct a probabilistic model that maps hyperparameters to the likelihood of achieving a certain score on the objective function. It constructs a probabilistic model of the objective function based on the observed evaluations and uses this model to guide the search process. By iteratively selecting promising hyperparameter configurations based on an acquisition function, Bayesian optimization gradually explores the hyperparameter space and converges towards the optimal solution. Bayesian optimization can explore a larger search space compared to more traditional hyperparameter optimization techniques like grid search and random search, thereby achieving more effective results in relatively shorter time frames.

In this work, one of the Python libraries called Hyperopt (Bergstra et al., 2015) is used to perform Bayesian optimization. Hyperopt is a library for Bayesian optimization that can implement the Tree-structured Parzen Estimator (TPE), which is more advanced than other optimization algorithms. There are four components in Bayesian optimization:

1. Objective function (F(x)): The function that one aims to minimize.

2. Domain space (X): The range of parameter values over which the objective is minimized.

3. Hyperparameter optimization function (TPE): This function creates the surrogate function and selects the next values to assess.

4. Trials: Each instance where the objective function is evaluated, recording the score and parameter pairs.

5. Max_eval: maximum evaluation number.

TPE is a bayesian-based approach that tries to build a probabilistic model. TPE implies that hyperparameter space exhibits a tree-like structure: the selection of a value for one hyperparameter determines the subsequent selection of another hyperparameter and the range of values available for it. TPE algorithm works as follows:

1. Create a randomly chosen initial point from domain space X: x∗.

2. Compute F(x∗). (The function F corresponds to the objective function, which in our case is the negative of accuracy.)

3. Utilizing the trial history, construct the conditional probability model P(F |x).

4. Select based on P(F |x), anticipating an improvement in F(x∗).

5. Calculate the actual value of F(x∗).

6. Iterate through steps 3-5 until one of the stopping criteria is met, such as i >max_eval. The goal is to find the global minimum of F(x) over X.

Table 3 displays the hyperparameter ranges for various algorithms, encompassing the newly proposed ABC_ANN method alongside other comparative techniques.

Table 3 Hyperparameter ranges of Bayesian optimization based on different classification algorithms.

Model	Parameters	Range	
	learning rate	[10e−8.10e−1]	
	hidden size 1	[100,150] and [25,35]	
	hidden size 2	[30,100] and [10,25]	
DAE	dropout rate 1	[0,0.3]	
	dropout rate 2	[0,0.3]	
	batch size	[1,1024]	
	epochs	[1,100]	
	act. func.	{tanh,sigmoid}	
	no. of particles	[3,20]	
	c1	[ 0.5,3]	
	c2	[0.5,3]	
PSO_ANN	no. of iter.	[5,100]	
	w	[0.1,2]	
	hidden size	[5,100]	
	act. func.	{tanh,sigmoid,relu}	
	no. of solutions	[10,20]	
	no. of generations	[ 10,100]	
GA_ANN	hidden size	[3,20]	
	no. of parents mating	[1,10]	
	act. func.	{tanh,sigmoid,relu}	
	batch size	{ 8,16,32,64,128,256 }	
	epoch	[50,200]	
	hidden size	[3,50]	
SGD_ANN	dropout rate	[0,0.4]	
	learning rate	[0.01,0.5]	
	momentum	[0.1,1]	
	act. func.	{tanh,sigmoid,relu}	
	batch size	{ 8,16,32,64,128,256 }	
	epoch	[50,200]	
	hidden size	[3,50]	
Adam_ANN	dropout rate	[0,0.4]	
	learning rate	[0.01,0.5]	
	act. func.	{tanh,sigmoid,relu}	
	HLS	[2,20]	
	lb	[−30,0]	
	ub	[0,30]	
Proposed ABC_ANN	evaluation number	[10000,120000]	
	limit	[10,200 ]	
	P	[10,200]	
	MR	[0.01,0.2]	
	threshold	[0.2,0.8]	
	act. func.	{tanh,sigmoid}	

Results and Discussion

Datasets and data preprocessing

This study utilizes the UNSW-NB15 and the NF-UNSW-NB15-v2 datasets to build the NIDS model. The UNSW-NB15 dataset consists of a combination of actual modern normal network activities and synthesized current network attack activities. It serves as an alternative to older benchmark datasets and is widely adopted for evaluating the performance of Network Intrusion Detection Systems (NIDS).

The NetFlow-based format of the UNSW-NB15 dataset, referred to as the NF-UNSW-NB15-v2 (Sarhan, Layeghy & Portmann, 2022), has been expanded with supplementary NetFlow attributes and labeled with corresponding attack categories. The dataset comprises a total of 43 features and 2,390,275 data flows, with 95,053 classified as attack samples and 2,295,222 as benign. To avoid and mitigate bias in model training, as part of the data pre-processing procedure, six attributes are removed from the dataset. These include minimum or maximum traffic Time to Live (TTLs), port numbers, IPv4 source, and destination addresses that do not significantly contribute to the classification performance and are highly correlated with class labels. Since this dataset does not provide users with pre-existing training and test sets, the experiments split the data into partitions containing an equal proportion of samples from both the benign and attack classes. Specifically, 33% of the data is allocated for testing, while the remaining portion constitutes the training set.

The training set of the UNSW-NB15 dataset contains 175,341 samples with 45 features. Among these samples, 56,000 are labeled as “normal” traffic, while 119,341 samples are labeled as “abnormal” traffic, representing various types of attacks. The testing set consists of 82,332 samples with the same feature size as the training set. Within the testing set, 37,000 samples are categorized as “normal” traffic, and the remaining 45,332 samples represent “abnormal” traffic with different attack types, including DoS, backdoors, generic attacks, analysis attacks, exploits, shell code, fuzzers, reconnaissance, and worms. UNSW-NB15 dataset contains categorical features. A category encoder technique is utilized to convert these categories into numerical values. Specifically, “service”, “state”, and “proto” are the three categorical features found in the dataset. Following the encoding process, the total number of features expands from 45 to 197.

To minimize the impact of different scales across features and to reduce computational costs and training time, normalization techniques are applied to both datasets. Several normalization strategies exist in the literature, such as the Max-Abs scaler, the Standard scaler, and the Min-Max scaler. Considering the sparsity analysis of both datasets, which indicates a significant proportion of zeros, the Max-abs scaler is found to be appropriate for these datasets. The Max-abs normalization technique scales each feature by its maximum absolute value while preserving all zeros. This normalization approach is used to scale all values in the dataset to the range of [0, 1]. By applying the Max-abs scaler, datasets are prepared for the following processing and analysis in the NIDS model construction.

Evaluation metrics

Evaluation metrics play an important role in evaluating the performance of machine learning algorithms. In addition to accuracy, it is important to take into account the F1-score, the FAR and the DR for a comprehensive understanding of the model’s performance. A confusion matrix (as shown in Table 4) is a table that summarizes the performance of a classification model. It provides a detailed breakdown of correct and incorrect predictions made by the model across different classes (Rainio, Teuho & Klén, 2024). Evaluation metrics and confusion matrix play crucial roles in assessing the performance of machine learning models, providing insights into their predictive capabilities and helping in the refinement and optimization of models for better performance.

Table 4 Confusion matrix.

	Predicted normal	Predicted abnormal	
Actual normal	True negative (TN)	False positive (FP)	
Actual abnormal	False negative (FN)	True positive (TP)	

In the context of the proposed network anomaly detection system, we define the performance metrics shown in Table 4 as follows:

• True positive (TP): The number of instances where the IDS model correctly identifies network anomalies.

• True negative (TN): The number of instances where the system correctly identifies normal network traffic or behavior, meaning no anomalies were present and none were detected.

• False positive (FP): The number of instances where normal network traffic is incorrectly flagged as anomalous by the IDS model.

• False negative (FN): The number of instances where actual network anomalies were present but not detected by the IDS model.

Some common evaluation metrics derived from the confusion matrix include accuracy, precision, recall, F1-score, DR, and FAR. The definition of accuracy is the ratio of correctly estimated samples to all samples (Eq. (5)). It provides a basic performance measure but may not be sufficient, especially in the case of unbalanced datasets. As is often the case with problems with network intrusion detection, imbalance occurs when there is an insufficient number from one class (eg. abnormal). In such cases, the importance of metrics such as the F1-score becomes even more apparent. The F1-score is a statistical measure of precision and recall (Eq. (8)). Precision measures the ratio of true positives (correctly predicted abnormal samples) to the total number of predicted positives, while recall measures the ratio of actual positives to the overall actual positive number. The F1-score is the harmonic mean of precision and recall, providing a balanced measure of accuracy, taking into account both false positives and false negatives.

DR, also known as true positive rate (TPR) or Sensitivity, is the ratio of true positive samples to the total number of actual positive samples (abnormal samples in the dataset) (Eq. (6)). It indicates the ability of the model to correctly identify positive instances. FAR, also known as false positive rate (FPR), is the ratio of false positive samples to the total number of actual negative samples (normal samples in the dataset) (Eq. (7)). It represents the proportion of normal instances that are incorrectly classified as abnormal.

By considering these evaluation metrics, including F1-score, DR, and FAR, alongside accuracy, a more comprehensive assessment of the model’s performance can be obtained, particularly in the presence of imbalanced datasets. In such cases, the focus is not only on overall accuracy but also on correctly identifying abnormal instances while minimizing false alarms.

(5) Accuracy=TP+TNTN+FP+FN+TP

(6) DetectionRateDRorTPR=TPFN+TP

(7) FalseAlarmRateFPR=FPTN+FP

(8) F1score=2∗TP2∗TP+FP+FN.

Experimental setup

The proposed methods were implemented utilizing the Colab platform offered by Google, which provides access to NVIDIA T4 GPUs. The GA, PSO and ANN with SGD and Adam optimization algorithms were conducted using PyGAD (Gad, 2021), pyswarms (Miranda, 2018), and Tensorflow (Abadi et al., 2015) libraries, respectively.

Experimental results

The experimental setup of this study encompasses three primary processes.

The first objective is to evaluate the contribution of newly extracted features that are obtained via Deep Autoencoder (DAE), to the classification task. These features were derived utilizing a deep learning technique, and their effect on the classification outcomes was assessed.

Secondly, an exploration into the impact of the number of selected features on the classification performance is conducted using XGBoost with five-fold cross-validation, which ensures the selection of the most appropriate features. This investigation entails applying the proposed DAE-based ABC-ANN algorithm with default hyperparameter settings to three distinct feature sets:

• Original features

• Encoded features obtained from the DAE

• Concatenation of the original and encoded features

To ascertain the optimal number of features, we employ a five-fold cross-validation approach, augmented by XGBoost feature selection. The objective is to systematically vary the number of selected features and evaluate their impact on classification metrics, including accuracy, F1-score, DR, and FAR.

During this procedure, a total of 30 features are selected from the UNSW-NB15 dataset (shown in Table 5), and 40 features are selected from the NF-UNSW-NB15-v2 dataset (shown in Table 6) based on the five-fold cross-validation XGBoost feature selection technique, as detailed in ‘Feature Selection via Extreme Gradient Boosting (XGBoost) Algorithm’.

Table 5 Selected 30 features using the five-fold cross-validation XGBoost method obtained from a combination of the UNSW-NB15 original features and encoded features.

The sum of all importance scores equals 1.

Feature name	Format	Feature importance scores	
sttl	integer	0.37267	
ct_srv_dst	integer	0.06874	
ct_dst_src_ltm	integer	0.04268	
encoded f42	float	0.03717	
synack	float	0.02834	
sbytes	integer	0.02769	
ct_state_ttl	integer	0.02617	
encoded f16	float	0.02571	
service=-	categorical	0.02395	
ct_srv_src	integer	0.02207	
ct_dst_sport_ltm	integer	0.019221	
encoded f43	float	0.01834	
encoded f15	float	0.01737	
smean	integer	0.01649	
encoded f48	float	0.01343	
proto=tcp	categorical	0.01192	
encoded f24	float	0.01173	
encoded f52	float	0.01146	
encoded f49	float	0.01140	
dbytes	integer	0.01131	
dmean	integer	0.00886	
encoded f18	float	0.00838	
encoded f27	float	0.00788	
encoded f26	float	0.00782	
service=http	categorical	0.00700	
encoded f21	float	0.00691	
service=dns	categorical	0.00642	
service=ftp	categorical	0.00619	
encoded f32	float	0.00482	
encoded f40	float	0.00482	

Table 6 Selected 40 features using the five-fold cross-validation XGBoost method obtained from a combination of the NF-UNSW-NB15-v2 original features and encoded features.

The sum of all importance scores equals 1.

Feature name	Format	Feature importance scores	
MIN_IP_PKT_LEN	integer	0.67513	
TCP_WIN_MAX_IN	integer	0.18188	
SHORTEST_FLOW_PKT	integer	0.02189	
DNS_QUERY_TYPE	integer	0.01588	
LONGEST_FLOW_PKT	integer	0.01269	
encoded f29	float	0.00533	
RETRANSMITTED_OUT_BYTES	integer	0.00418	
SERVER_TCP_FLAGS	integer	0.00395	
L7_PROTO	float	0.00391	
PROTOCOL	integer	0.00390	
encoded f21	float	0.00380	
encoded f22	float	0.00363	
TCP_FLAGS	integer	0.00359	
OUT_BYTES	integer	0.00296	
encoded f17	float	0.00289	
NUM_PKTS_UP_TO_128_BYTES	integer	0.00270	
RETRANSMITTED_OUT_PKTS	integer	0.00249	
encoded f13	float	0.00223	
OUT_PKTS	integer	0.00220	
encoded f5	float	0.00219	
FTP_COMMAND_RET_CODE	integer	0.00205	
CLIENT_TCP_FLAGS	integer	0.00192	
DST_TO_SRC_SECOND_BYTES	float	0.00190	
SRC_TO_DST_AVG_THROUGHPUT	integer	0.00182	
encoded f12	float	0.00172	
IN_PKTS	integer	0.00170	
TCP_WIN_MAX_OUT	integer	0.00164	
encoded f10	float	0.00139	
encoded f4	float	0.00137	
DST_TO_SRC_AVG_THROUGHPUT	integer	0.00133	
encoded f14	float	0.00132	
NUM_PKTS_128_TO_256_BYTES	integer	0.00127	
SRC_TO_DST_SECOND_BYTES	integer	0.00126	
RETRANSMITTED_IN_BYTES	integer	0.00124	
encoded f9	float	0.00121	
encoded f3	float	0.00121	
IN_BYTES	integer	0.00116	
encoded f8	float	0.00110	
encoded f18	float	0.00107	
encoded f11	float	0.00107	

Ablation studies were conducted with the aim of evaluating the individual contributions of each process, including feature extraction, feature selection, and the effects of parallelization and vectorization on training times. Figs. 4 and 5 present the accuracy and F1 results derived from various configurations for the UNSW-NB15 dataset. These configurations include:

Figure 4 Accuracy of the classification model based on the different number of feature subsets obtained from the five-fold cross validation XGBoost algorithm, applied on the UNSW-NB15 dataset.

Figure 5 F1-scores of the classification model based on the different number of feature subsets obtained from the five-fold cross validation XGBoost algorithm, applied on the UNSW-NB15 dataset.

• Without using feature extraction and selection, which generates accuracy and F1-scores of 0.81 and 0.86, respectively.

• Solely using extracted encoded features, yielding accuracy and F1-scores of 0.81 and 0.85, respectively.

• The accuracy and F1-scores, that are obtained using different feature sets.

Similarly, Figs. 6 and 7 encompass the accuracy and F1 results derived from various configurations for the NF-UNSW-NB15-v2 dataset. These configurations include:

Figure 6 Accuracy of the classification model based on the different number of feature subsets obtained from the five-fold cross validation XGBoost algorithm, applied on the NF_UNSW-NB15_v2 dataset.

Figure 7 F1-scores of the classification model based on the different number of feature subsets obtained from the five-fold cross validation XGBoost algorithm, applied on the NF_UNSW-NB15_v2 dataset.

• Without using feature extraction and selection, which generates accuracy and F1-scores of 0.98 and 0.79, respectively.

• Solely using extracted encoded features, yielding accuracy and F1-scores of 0.99 and 0.81, respectively.

• The accuracy and F1-scores are obtained using different feature sets.

Thirdly, the performance of the proposed DAE-based ABC-ANN method has been compared with benchmark metaheuristics, namely the Genetic Algorithm (GA) and particle swarm algorithm (PSO). Furthermore, we have compared the performance of the proposed DAE based ABC-ANN method with a conventional ANN approach that involves error back propagation and weight adjustment using a Stochastic Gradient Descent (SGD) and Adam optimization algorithms. In this study, with the goal of minimizing computational load, all matrices referenced in the Algorithm 1  were vectorized utilizing the python programming language and numpy library, as opposed to Python lists and loops (detailed explanation can be found in ‘Data Vectorization and Parallel Computation on GPU’). This represents a significant contribution to the literature. Thus, the computational load is substantially reduced to a minimum. In consideration of the big datasets as in this study, it became crucial to accelerate the model. In order to accomplish this, GPU parallelization has also been utilized for vectorized loops, which significantly increased the overall speed and efficiency. The utilization of CuPy (Nishino & Loomis, 2017), an open-source library specifically developed for accelerating matrix operations on NVIDIA GPUs, enabled this acceleration.

All aforementioned classification algorithms were applied to the optimal set of 30 and 40 features selected in the previous step using XGBoost feature selection.

Table 7 presents the optimal parameters achieved after 150 iterations using the Bayesian optimization algorithm. In order to demonstrate the effectiveness of the Bayesian optimization algorithm, its performances were compared with the randomized search algorithm. It can be seen in Table 8 that although the randomized search algorithm was run for 250 iterations, it could not pass the Bayesian optimization algorithm in terms of evaluation metrics including F1-score, accuracy, DR, and FAR. After Bayesian optimization has been conducted on all classification algorithms with 150 iterations, the optimum results of this comparison are presented in Tables 9 and 10. The experimental results demonstrate a significant improvement in network intrusion detection with the proposed approach. DR increased from 0.76 to 0.81, and FAR decreased from 0.0016 to 0.005 when compared to the ANN-BP algorithm on the UNSW-NB15 dataset. Additionally, FAR decreased from 0.006 to 0.0003 compared to the ANN-BP algorithm on the NF-UNSW-NB15-v2 dataset. It is observed that the test results of PSO and GA on the NF-UNSW-NB15-v2 dataset are not satisfactory, indicating that they require more time and hardware resources to reach the optimum. These findings highlight the effectiveness of our proposed approach in enhancing network security against intrusions.

Table 7 The optimal parameters found by the Bayesian hyperparameter optimization algorithm on the UNSW-NB15 and NF-UNSW-NB15-v2 datasets.

Model	Parameters	Opt.values UNSW-NB15	Opt.values NF-UNSW-NB15-v2	
DAE	learning rate hidden size 1 hidden size 2 dropout 1 dropout 2 batch size epochs act. func.	0.3 100 53 0.1 0.0 256 90 sigmoid	0.24 31 25 0.1 0.1 37 12 sigmoid	
PSO_ANN	no. of particles c1 c2 w hidden size act. func. no. of iter.	11 1.36 1.88 0.38 3 sigmoid 49	8 1.25 2.3 0.48 3 sigmoid 37	
GA_ANN	no. of solutions no. of generations hidden size no. of parents mating act. func.	14 24 3 8 tanh	6 1 9 2 tanh	
SGD_ANN	batch size epoch hidden size dropout rate learning rate momentum act. func.	16 108 35 0.1 0.08 0.35 relu	128 190 42 0.3 0.003 0.02 tanh	
Adam_ANN	batch size epoch hidden size dropout rate learning rate act. func.	32 88 25 0 0.17 sigmoid	128 113 15 0.1 0.058 relu	
proposed ABC_ANN	HLS lb ub evaluation number limit P MR threshold act. func.	3 -20 20 60,008 50 40 0.054 0.5 sigmoid	4 -14.6 13.8 58,567 69 68 0.04 0.5 sigmoid	

Table 8 The best performance evaluation results of the UNSW-NB15 dataset with 30 selected features and the NF-UNSW-NB15-v2 dataset with 20 selected features, calculated using the Bayesian hyperparameter optimization algorithm with 150 iterations and the randomized search strategy with 250 iterations.

Dataset	Model	Optimization strategy	No. of iterations	Accuracy	F1	DR	FPR	TTime	
UNSW-NB15	proposed ABC_ANN (GPU)	Randomized Search Bayesian Optimization	250 150	0.82 0.86	0.86 0.88	0.75 0.81	0.008 0.005	6 min 16 s 3 min 23 s	
NF-UNSW- NB15-v2	proposed ABC_ANN (GPU)	Randomized Search Bayesian Optimization	250 150	0.99 0.99	0.84 0.89	0.73 0.81	0.0008 0.0003	9 min 58 s 8 min 41 s	
Notes.

TTime, Training Time.

Table 9 The best performance evaluation results of the UNSW-NB15 dataset with 30 selected features, calculated using the Bayesian hyperparameter optimization algorithm after 150 iterations.

Model	Accuracy	F1	DR	FPR	Training time	
GA_ANN	0.75	0.81	0.72	0.15	54 min 23 s	
PSO_ANN	0.81	0.85	0.74	0.006	34 min 31 s	
SGD_ANN	0.82	0.86	0.76	0.016	7 min 25 s	
Adam_ANN	0.84	0.87	0.79	0.032	4 min 14 s	
proposed ABC_ANN (CPU)	0.86	0.88	0.81	0.005	23 min 47 s	
proposed ABC_ANN (GPU)	0.86	0.88	0.81	0.005	3 min 23 s	

Table 10 The best performance evaluation results of the NF-UNSW-NB15-v2 dataset with 40 selected features, calculated using the Bayesian hyperparameter optimization algorithm after 150 iterations.

Model	Accuracy	F1	DR	FPR	Training time	
GA_ANN	0.96	0.02	0.36	0.035	2 hr 38 min	
PSO_ANN	0.96	0.0013	0.002	0.036	4 hr 40 min	
SGD_ANN	0.99	0.86	0.875	0.006	50 min 42 s	
Adam_ANN	0.99	0.87	0.84	0.003	9 min 28 s	
proposed ABC_ANN (CPU)	0.99	0.89	0.81	0.0003	2 hr 16 min	
proposed ABC_ANN (GPU)	0.99	0.89	0.81	0.0003	8 min 41 s	

Furthermore, in order to ensure the reliability of the results, by executing the models 20 times for the UNSW-NB15 and the NF-UNSW-NB15-v2 datasets, the best, worst, average training time, and standard deviation values were recorded for each classifier. Table 11 summarizes the best, worst, average, and standard deviation values obtained after repeating the best model achieved in the UNSW-NB15 dataset 20 times. Table 12, on the other hand, provides a summary of the same results obtained for the NF-UNSW-NB15-v2 dataset.

Table 11 The time in seconds required to train each classifier on the UNSW-NB15 dataset.

Model	Best time	Worst time	Avg time	Std.	
GA_ANN	5,454	6,169	5,698.68	251.96	
PSO_ANN	4,326	4,499	4,407.37	51.78	
SGD_ANN	272	791	431.63	142.05	
Adam_ANN	209	552	316.63	79.77	
Proposed ABC_ANN (CPU)	1,412	1,533	1,442.42	34.32	
Proposed ABC_ANN (GPU)	144	176	146.68	7.14	

Table 12 The time in seconds required to train each classifier on the NF-UNSW-NB15-v2 dataset.

Model	Best time	Worst time	Avg time	Std.	
GA_ANN	8,760	9,507	9,080.8	339.386	
PSO_ANN	16,816	17,625	17,206	433.615	
SGD_ANN	2,951	4,394	4,519.33	544.72	
Adam_ANN	519	857	636.25	104.549	
Proposed ABC_ANN (CPU)	8,040	8,160	8,086	49.98	
Proposed ABC_ANN (GPU)	504	509	506.94	1.545	

Overall, the experimental setup has involved evaluating the contribution of DAE-extracted features, exploring the influence of the number of selected features, comparing the performance of the proposed hybrid DAE-based ABC-ANN method with benchmark metaheuristics, and contrasting it with conventional ANN approaches with SGD and Adam optimization using the sklearn TensorFlow library.

The results demonstrate that the proposed hybrid DAE-based ABC-ANN approach outperforms state-of-the-art algorithms in terms of accuracy, F1-score, detection rate (DR), false positive rate (FPR), and training time on the UNSW-NB15 and NF-UNSW-NB15 datasets.

Conclusion

This study combines DAE with vectorized and GPU-parallelized ABC-ANN to efficiently address big data problems by searching for global solutions in a faster manner. While existing methods may achieve high accuracy, they may suffer from high training times, low detection rates, and computational complexity. In this study, the ABC algorithm has been vectorized and coded to run in parallel on GPUs to address these issues. Additionally, DAE and feature selection have been conducted to obtain a more robust dataset representation.

The proposed DAE-based ABC-ANN method is compared with the conventional ANN backpropagation (ANN-BP), ANN-PSO, ANN-GA and ANN-Adam optimization algorithms, and the results are thoroughly analyzed. The ABC algorithm in the ANN training phase allows for the avoidance of local minimum solutions by conducting a high-performance search in the solution space.

This study investigated the XGBoost algorithm for feature selection, and the DAE for feature extraction in conjunction with numerous approaches, including PSO, GA, SGD, and Adam optimization, to develop reliable, efficient, and accurate IDSs. In order to evaluate the effectiveness of these techniques, the benchmark UNSW-NB15 and up-to-date NF-UNSW-NB15-v2 datasets were trained and tested. Firstly, the DAE-based feature extraction method was conducted with bayesian hyperparameter optimization on datasets in order to extract the most representative features, and it resulted in 53 encoded features on the UNSW-NB15 and 24 encoded features on the NF-UNSW-NB15-v2 datasets. Secondly, the XGBoost-based feature selection method was used to select the best features from the combination of original and encoded features. Thirdly, an ABC-ANN is proposed with CPU and GPU parallelization, which allows the use of ABC intelligence in big data problems. The computational costs of the proposed ANN-ABC method impose limitations on the GPU. Lastly, a Bayesian-based hyperparameter optimization technique is conducted on all experimental algorithms and the proposed ABC-ANN algorithm in order to find the best hyperparameter combinations that improve detection accuracy and F1 score.

To place our findings in context, this study has conducted a comprehensive literature analysis. In addition, it has created a summary of the performance results acquired by the various algorithms and compared them using the proposed method. Consequently, the results have demonstrated clearly that the proposed DAE-based ABC-ANN method is superior to alternative approaches across all evaluation metrics mentioned in ‘Evaluation Metrics’. The experimental results reveal a notable improvement in network intrusion detection through this proposed approach, exhibiting an increase in DR by 0.76 to 0.81 and a reduction in FAR by 0.016 to 0.005 compared to the ANN-BP algorithm on the UNSW-NB15 dataset (Table 9). Furthermore, there is a reduction in FAR by 0.006 to 0.0003 compared to the ANN-BP algorithm on the NF-UNSW-NB15-v2 dataset (Table 10). These findings underscore the effectiveness of our proposed approach in enhancing network security against network intrusions.

However, there are still certain aspects of the DAE-based ABC-ANN that need improvement. Our proposed approach surpasses conventional methods in computational efficiency and classification metrics by leveraging the proposed optimization technique. Nevertheless, the model encounters limitations in acceleration due to insufficient hardware, notwithstanding its high hardware requirements. Exploring additional hardware resources may lead to improved results. Future work includes the investigation of hybrid models to improve the anomaly detection performance. By incorporating the ABC algorithm for tuning hyperparameters in the proposed method, we can simultaneously optimize parameters and ANN weights. This dual optimization process contributes to the enhancement of the overall methodology.

Supplemental Information

Supplemental Information 1 Proposed ABC-ANN code

Supplemental Information 2 Justification for added co-author

Additional Information and Declarations

Competing Interests

Author Contributions

Data Availability

The authors declare there are no competing interests.

Hilal Hacılar conceived and designed the experiments, performed the experiments, analyzed the data, performed the computation work, prepared figures and/or tables, authored or reviewed drafts of the article, and approved the final draft.

Bilge Kagan Dedeturk conceived and designed the experiments, performed the experiments, performed the computation work, authored or reviewed drafts of the article, and approved the final draft.

Burcu Bakir-Gungor conceived and designed the experiments, performed the experiments, authored or reviewed drafts of the article, and approved the final draft.

Vehbi Cagri Gungor conceived and designed the experiments, performed the experiments, authored or reviewed drafts of the article, and approved the final draft.

The following information was supplied regarding data availability:

The NF-UNSW-NB15-V2 dataset is publicly available at: Moustafa, Nour, and Jill Slay. “UNSW-NB15: a comprehensive data set for network intrusion detection systems (UNSW-NB15 network data set).” Military Communications and Information Systems Conference (MilCIS), 2015. IEEE, 2015. https://research.unsw.edu.au/projects/unsw-nb15-dataset.

The NF-UNSW-NB15-V2 dataset is also available at Kaggle: https://www.kaggle.com/datasets/dhoogla/nfunswnb15v2.

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
