# Peer review of "Network anomaly detection using Deep Autoencoder and parallel Artificial Bee Colony algorithm-trained neural network"

_PeerJ Computer Science, doi:10.7717/peerj-cs.2333_

## Round 0.1 · original submission · Major Revisions

The paper needs to follow the reviewer's comments.

Reviewer 3 has suggested that you cite specific references. You are welcome to add it/them if you believe they are relevant. However, you are not required to include these citations, and if you do not include them, this will not influence my decision.

**Language Note:** The review process has identified that the English language must be improved. PeerJ can provide language editing services - please contact us at [email protected] for pricing (be sure to provide your manuscript number and title). Alternatively, you should make your own arrangements to improve the language quality and provide details in your response letter. – PeerJ Staff

·

Basic reporting

No comment.

Experimental design

no comment

Validity of the findings

no comment

Additional comments

Dataset authers used is about ten years ago, and there are already newer dataset, such as CSE-CIC-IDS2018. Please explain clearly about reasons of choosing UNSW-NB15 or add newer dataset to experiment and evaluate.

Reviewer 2 ·

Basic reporting

Significant improvements are needed in the use of the English language. For instance, the sentence "Traditional training algorithms like back-progration may get stuck in local minima" does not adhere to appropriate scientific English. An example of proficient scientific expression would be, "Conventional training algorithms, such as backpropagation, may encounter challenges becoming ensnared in local minima during the optimization process."

There appears to be a mix of references and tables in this work.

The absence of motivational elements and elucidation of research challenges is conspicuous in this study. It is perplexing how the authors approach their work without a comprehensive discussion of the motivation and challenges inherent in the research. It is strongly advised that the authors incorporate a dedicated sub-section to clearly articulate the motivation and present research questions, in addition to outlining their research design.

The paper exhibits a notable lack of organization, with the proposed method improperly positioned preceding the evaluation matrices. Feature selection should immediately follow the discussion of the dataset. The absence of a discussion on feature normalization is noteworthy, and Figure 2 is incorrectly placed within the document.

A mathematical illustration of Bayesian Optimization in consideration of the selected variables is notably absent. Furthermore, the rationale behind explaining data vectorization instead of data normalization is unclear and warrants clarification from the authors.

Why do authors use DAE? When sparse AE is more efficient due to its hidden layer containing fewer nodes than the number of inputs lies.

Experimental design

The choice of employing outdated data, specifically UNSW-NB15, appears questionable, mainly when more recent datasets such as IoT-23, LITNET-2020, and UNSW_NB15 V2 are easily accessible. Opting for conventional data is an imprudent decision, given the availability of contemporary alternatives.

The experimental section suffers from a significant lack of organization, rendering it challenging to comprehend. Furthermore, the absence of any discussion on the obtained results is doubtful. Additionally, there is a notable deficiency in the disclosure of experimental outcomes. The current manuscript fails to contribute any innovative elements to the experimental design, lacking novelty in its approach.

Validity of the findings

The validation of the findings is impeded by the disorganized evidence in the presentation of experimental results, compounded by the absence of essential complements to support the experimental outcomes. The inclusion of merely two figures falls short of providing comprehensive insights, with a notable omission of ablation studies or comparative analyses for this study.

Regrettably, this paper lacks novelty and fails to elucidate any substantial scientific contributions.

Additional comments

The authors are strongly encouraged to revise the manuscript meticulously, with a particular emphasis on enhancing the paper's overall organization.

It is recommended that the authors peruse relevant literature papers within this domain to observe how such works are presented and organized. This practice will provide valuable insights into effective approaches employed by other scholars in structuring their contributions.

Reviewer 3 ·

Basic reporting

In this study, Artificial Neural Networks (ANNs), a type of machine learning algorithm, are used to find optimal network weight sets during training. Traditional training algorithms, such as backpropagation, may face challenges like getting stuck in local minima, slow global search strategies, and low detection rates. To overcome these issues, the study proposes a deep autoencoder (DAE)-based, vectorized, and parallelized Artificial Bee Colony (ABC) algorithm for training feed-forward artificial neural networks on the UNSW-NB15 dataset. The experimental results demonstrate a significant improvement in network intrusion detection, with the proposed DAE-based ABC-ANN showing the best and most competitive performance. Specifically, there is an increase in Detection Rate (DR) by 0.76 to 0.81 and a reduction in False Alarm Rate (FAR) by 0.0016 to 0.005 compared to the ANN-BP algorithm. The flow of the paper is nice, yet there are some issues to be addressed to enhance the paper.

Strength:
1) Motivation is good and problem is well articulated.

2) Both theoretical and experimental analysis are given.

Experimental design

Suggestion:
1) Lack of citations in some significant places. Like section Metahuristics on NIDS,Evaluation Metrics and so on.

Graph neural network-based anomaly detection in multivariate time series[C]//Proceedings of the AAAI conference on artificial intelligence. 2021, 35(5): 4027-4035
HELAD: A novel network anomaly detection model based on heterogeneous ensemble learning[J]. Computer Networks, 2020, 169: 107049

2) Considering a single dataset, it is advisable for the authors to incorporate references to advanced network intrusion detection approaches, provide an overview of commonly employed datasets, and demonstrate the effectiveness of their scheme from diverse perspectives (MAWILab, Yelp , PubMed, IDS and so on).

Graph neural network-based anomaly detection in multivariate time series[C]//Proceedings of the AAAI conference on artificial intelligence. 2021, 35(5): 4027-4035

A K-Means clustering and SVM based hybrid concept drift detection technique for network anomaly detection[J]. Expert Systems with Applications, 2022, 193: 116510

The MVTec anomaly detection dataset: a comprehensive real-world dataset for unsupervised anomaly detection[J]. International Journal of Computer Vision, 2021, 129(4): 1038-1059.

3) I suggest the authors to explicitly define the threat model, that is, what are the capabilities allowed to an attacker. Otherwise it is difficult to assess the security of the proposed protocol.

"Quantum2FA: Efficient Quantum-Resistant Two-Factor Authentication Scheme for Mobile Devices", IEEE Transactions on Dependable and Secure Computing, 2023
"Two Birds with One Stone: Two-Factor Authentication with Security Beyond Conventional Bound". IEEE Transactions on Dependable and Secure Computing, 2018.
Fuzzy logic-based DDoS attacks and network traffic anomaly detection methods: Classification, overview, and future perspectives[J]. Information Sciences, 2023.

4) The author employed the Deep Autoencoder, Artificial Bee Colony, and various other algorithms for network intrusion detection. Please provide an overview of each method within the system illustration, detailing the role played by each module to underscore the distinctiveness of the proposed scheme. Describe the roles and functions of Algorithms 1-7 in conjunction with a system diagram

Few-shot network anomaly detection via cross-network meta-learning[C]//Proceedings of the Web Conference 2021. 2021: 2448-2456.
Mgfn: Magnitude-contrastive glance-and-focus network for weakly-supervised video anomaly detection[C]//Proceedings of the AAAI Conference on Artificial Intelligence. 2023, 37(1): 387-395.

5) Comparative experiments and evaluation criteria are too homogenous. The authors fail to establish the significance of their contributions in a convincing manner. It is recommended that the authors refer to the latest at least five state-of-the-art works to refine the scheme.

Deep learning for anomaly detection: A review[J]. ACM computing surveys (CSUR), 2021, 54(2): 1-38.
Network anomaly detection based on selective ensemble algorithm[J]. The Journal of Supercomputing, 2021, 77: 2875-2896.
Future frame prediction network for video anomaly detection[J]. IEEE transactions on pattern analysis and machine intelligence, 2021, 44(11): 7505-7520.
An efficient network behavior anomaly detection using a hybrid DBN-LSTM network[J]. Computers & Security, 2022, 114: 102600.
Few-shot network anomaly detection via cross-network meta-learning[C]//Proceedings of the Web Conference 2021. 2021: 2448-2456.


6)The writing can be improved. It is recommended that the authors standardize and modify the font of the formulas.
a. Please provide a comprehensive model of the system, supporting theoretical evidence, a detailed description of the experimental setup, and relevant captions.

b. Please review the text for grammatical errors and make necessary corrections for improvement.

Validity of the findings

I would suggest a minor revision to this paper.

---

## Round 0.2 · Minor Revisions

Please carefully consider the required modifications that reviewers reported

Reviewer 2 ·

Basic reporting

No comments

Experimental design

No comments

Validity of the findings

No comments

Additional comments

I would encourage the authors to clearly address the TP, TN, FP, and FN, considering your work. Don't use the term just like TN- True Negative, and so on. It doesn't make sense. You must disclose the true negative in terms of what?

Reviewer 3 ·

Basic reporting

The revised version has addressed most of my concerns, and I still have one question.

1) I suggest the authors to explicitly define the threat model, that is, what are the capabilities allowed to an attacker. Otherwise it is difficult to assess the security of the proposed protocol. Currently, the authors list some attack vectors in Section 3 "PROPOSED METHOD" but do not provide the necessary references.
Particularly, as shown in Figure 1, there may be a number of entities involved. Is there some form of (user) authentication needed? As Asset Movement applications are security-critical, are the following multi-factor authentication schemes suitable? The authors may briefly discuss this.

Experimental design

It is OK.

Validity of the findings

It is OK.

Additional comments

It is OK.

---

## Round 0.3 · accepted · Accept

Congratulations on having the required paper modifications and satisfying all the reviewers' comments.

Reviewer 3 ·

Basic reporting

The revised version has addressed most of my concerns, and I suggest an acceptance.

Experimental design

see bellow

Validity of the findings

see bellow

Additional comments

see bellow